# Protective Effect of Methyl Sulfonyl Methane on the Progression of Age-Induced Bone Loss by Regulating Oxidative Stress-Mediated Bone Resorption

**DOI:** 10.3390/antiox14020216

**Published:** 2025-02-13

**Authors:** Duo Zhang, Leilei Wang, Lu Tang, Yeting Zhang, Huaiyong Zhang, Lin Zou

**Affiliations:** 1Physical Education Department, Civil Aviation Flight University of China, Guanghan 618307, China; summer@cafuc.edu.cn (D.Z.); lutangcafuc@163.com (L.T.); zhangyt@cafuc.edu.cn (Y.Z.); 2Key Laboratory of Animal Biochemistry and Nutrition of Agriculture Ministry, College of Animal Science and Technology, Henan Agricultural University, Zhengzhou 450046, China; wl911152@163.com; 3Laboratory for Animal Nutrition and Animal Product Quality, Department of Animal Sciences and Aquatic Ecology, Ghent University, 9000 Ghent, Belgium

**Keywords:** age-related bone loss, oxidative stress, MSM, bone resorption

## Abstract

Aging is associated with detrimental bone loss, often leading to fragility fractures, which may be driven by oxidative stress. In this study, the outcomes of comparing the differences among young, adult and aged C57BL/6J mice found that the trabecular bone volume was significantly lower in the aged mice compared to young mice, and the bone characteristics were significantly correlated with the oxidative status. To counteract the adverse effects of aging, methyl sulfonyl methane (MSM), a stable metabolite of dimethyl sulfoxide, was used to supplement the drinking water (400 mg/kg/day) of the aged mice (73 weeks old) for 8 weeks. The MSM supplementation improved the maximum load, bone microarchitecture, and mRNA levels of osteocyte-specific genes in the tibia. Furthermore, MSM reduced the serum level of the C-terminal telopeptide of type I collagen, a marker of bone resorption, and downregulated the mRNA levels of genes related to osteoclast proliferation and activity. MSM also decreased the levels of pro-inflammatory cytokines in both the serum and bone marrow. Importantly, the MSM-treated mice exhibited an enhanced antioxidant status, characterized by increased glutathione peroxidase (GPx) activity and glutathione concentration in plasma, erythrocytes and bone marrow. These improvements were linked to the activation of the nuclear factor E2 related factor 2 (Nrf2) pathway and its downstream antioxidant gene expression, including that of superoxide dismutase and GPx. These findings suggested that age-related bone loss is closely tied to oxidative stress, and MSM supplementation effectively reverses bone loss by mitigating oxidative stress-mediated bone resorption.

## 1. Introduction

Aging is a natural, complex, and irreversible pathophysiological process characterized by a gradual decline in physiological functions, increasing the risk of diseases and mortality. A comprehensive understanding of the aging process and its mechanisms is crucial for developing therapeutic strategies for age-related diseases. Bone problems, commonly associated with aging, are often accompanied by osteoporosis, a condition marked by compromised bone microstructure and reduced bone mass [1]. This arises from an imbalance in bone remodeling, resulting in excessive bone resorption and impaired bone formation [2]. Notably, aging is frequently accompanied by oxidative stress due to disrupted balance between antioxidants and oxidants [3], leading to the overproduction of reactive oxygen species (ROS) [4]. Antioxidant defenses, comprising both endogenous and exogenous antioxidants such as catalase (CAT), superoxide dismutase (SOD), and glutathione peroxidase (GPx), protect tissues and organs from free radical damage [3]. Increasing evidence suggests that oxidative stress plays a critical role in disrupting bone mass homeostasis [5,6]. For example, SOD1 knockout mice exhibited a reduced bone turnover and low bone mass, and vitamin C supplementation could restore bone mass [7]. Additionally, age-related increases in osteocyte and osteoblast apoptosis, and decreases in bone remodeling, bone formation, bone mineral density, and bone strength have been linked to an elevated ROS level and diminished glutathione reductase (GR) activity; however, these effects could be mitigated by antioxidants such as N-acetyl-L-cysteine [8]. These findings suggest that age-related bone loss may be closely linked to increased oxidative stress.

Methyl sulfonyl methane (MSM), a stable metabolite of dimethyl sulfoxide, plays a role in the sulfur cycle and is widely present in both animals and humans. MSM is primarily absorbed in the small intestine and passively diffuses across cell membranes, including the blood–brain barrier [9]. Research has indicated that intestinal absorption of MSM was unsaturable and carrier-independent, with the absorbed compound being rapidly distributed to various tissues [10]. After administering a single oral dose of 500 mg/kg MSM to rats, the compound was rapidly absorbed within 2.1 h, exhibited a half-life of 12.2 h, and was predominantly excreted via urine (approximately 85.8%), with only 3% excreted through feces [11]. Previous evidence revealed that MSM demonstrated a wide range of biological functions, including antioxidant activity, free-radical scavenging, immune modulation, anti-inflammatory effects, alleviation of muscle soreness, antiallergic properties, and anticancer effects [12]. Specifically, MSM suppressed the production of pro-inflammatory mediators such as prostaglandin E2, nitric oxide (NO), interleukin-6 (IL-6), and tumor necrosis factor-α (TNF-α) by inhibiting the nuclear factor kappa B (NF-κB) signaling pathway [13]. Additionally, MSM mitigated monocrotaline-induced reduction in serum antioxidant status in pulmonary hypertensive rats [14]. Notably, MSM regulated the balance between antioxidant enzymes and ROS through transcription factors such as NF-κB, nuclear factor erythroid 2-related factor 2 (Nrf2), p53, and signal transducers and activators of transcription (STAT) [12]. Importantly, studies have shown that MSM supplementation promoted osteoblast differentiation by upregulating osteoblast- and osteogenic-specific markers, including runt-related transcription factor 2 (Runx2), alkaline phosphatase (ALP), osteocalcin (OCN), and bone sialoprotein (BSP) [15,16], suggesting that MSM supplementation may serve as an effective adjunctive treatment for age-related bone loss.

In this study, we hypothesized that oxidative stress contributes to age-induced bone loss and that MSM supplementation could mitigate this loss by modulating oxidative stress. This investigation aims to provide insights into the potential application of MSM in treating bone diseases associated with aging. To test this hypothesis, histological analysis, µCT scanning, and oxidative status assessments were conducted to explore the molecular mechanisms linking MSM to bone metabolism.

## 2. Materials and Methods

### 2.1. Animals and Diets

Three-week-old weanling C57BL/6J male mice were used in this trial. The mice were housed in a temperature- and humidity-controlled room with a standard 12 h light/dark cycle, maintained at 21–22 °C and 50–70% relative humidity. They had ad libitum access to water and a diet formulated to meet nutritional requirements based on the AIN-93 standard (Appendix A). All procedures adhered to the ethical guidelines of the Institutional Animal Care and Use Committee of Henan Agricultural University (Approval No. HNND2021-12).

### 2.2. Study Design

A total of 24 3-week-old weanling C57BL/6J male mice were housed in groups of 4 per cage and fed the same diet to investigate the age-related alterations in bone mass and antioxidant status. Body weight was recorded every two weeks. Mice were categorized as young (8 weeks), adult (32 weeks) and aged (72 weeks), with samples collected from each cage, according to Figure 1A. The left tibia and blood were harvested and stored at −80 °C for further analysis. To evaluate whether reducing oxidative stress could reverse age-related bone loss, MSM, an antioxidant reagent (Appendix A), was used in this study. After a one-week acclimatization period, 24 aged (73-week-old) C57BL/6J mice were divided into two groups and administered either 0 (Age) or 400 mg/kg/day of MSM (Aged + MSM) for 8 weeks, as shown in Appendix A. MSM, obtained from Bergstrom Nutrition (Vancouver, Washington, DC, USA), was dissolved in drinking water, and the dosage was based on a previous study [17]. All mice were fed the same AIN-93 diet. During the experimental period, body weight, feed intake and water consumption were recorded every four weeks (at 77 and 81 weeks of age). Blood, bone marrow, and proximal tibia samples were collected for further analysis.

### 2.3. Assessment of Bone Mass Using µCT-Scans

Bone mass was assessed using a µCT scanner equipped with an X-Ray WorX directional X-ray tube (X-RAY WorX GmbH, Garbsen, Germany) and a Perkin-Elmer flat-panel detector (PerkinElmer, Inc., Shelton, CT, USA) measuring 40 × 40 cm, with a pixel pitch of 200 µm [18]. The scanning parameters were as follows: a tube voltage of 130 kV, a beam power of 10 W, and no beam filtration. Each scan consisted of 2001 projections taken over a 360° rotation, with an exposure time of 1 s per projection. The resulting voxel size was 4 µm. Reconstruction and analysis of the 2D slice stack were performed using custom scripts in version 1.8 of ImageJ (National Institutes of Health, USA). Analyzed parameters included trabecular bone characteristics such as bone-volume-to-total-volume ratio (BV/TV, %), trabecular number (Tb.N, No./mm), thickness (Tb.Th, mm), separation (Tb.Sp, mm), and connectivity density (Tb.Cd, mm^3^). Cortical bone properties were also assessed, including cortical area (Ct.Ar, mm^2^), thickness (Ct.Th, mm), and bone mineral density (Ct.BMD, mgHA/cm^3^).

### 2.4. Mechanical Testing of Tibia

Following the µCT scan, the tibiae were subjected to a 3-point bending test to evaluate the mechanical properties of tibia. The loading point was applied to the mid-diaphysis of each tibia until bone failure occurred. Force–displacement data were recorded and used to calculate the whole bone stiffness (the slope of the linear portion of the load–displacement curve, N/mm) and the maximum force (max. load, Newtons) as described previously [19].

### 2.5. Osteoclast Frequency Determination

A part of the left tibia was harvested during dissection and fixed in 10% neutral-buffered formalin for 3 d, then decalcified in 14% ethylenediaminetetraacetic acid (EDTA) for 14 d. The samples were embedded following dehydration in 100% ethanol. The 10 μm thick frontal sections were stained with tartrate-resistant acid phosphatase (TRAP) (Sigma-Aldrich chemicals, St. Louis, MO, USA) to visualize the number of osteoclasts per bone perimeter (N.Oc/B.Pm) in accordance with the manufacturer’s instructions.

### 2.6. Serum Biochemical Markers of Bone Turnover

C-terminal telopeptide of type I collagen (CTx), a marker of bone resorption, was measured using an enzyme-linked immunosorbent assay (ELISA) kit (MyBioSource, Inc., San Diego, CA, USA). Biochemical markers of bone formation, including N-terminal propeptide of procollagen type I (P1NP) and osteocalcin (OCN), were quantified using corresponding ELISA kits obtained from Immunodiagnostic Systems (Boldon, UK) and Immutopics International (San Clemente, CA, USA), respectively.

### 2.7. Oxidative Stability Measures

Plasma, erythrocyte, and bone marrow measurements were employed to evaluate the alterations in oxidative stability due to aging and MSM treatment. Buffered aqueous extracts of bone marrow were prepared by mixing with 1% Triton X-100 phosphate buffer (pH 7; 50 mmol/L), followed by homogenization, centrifugation, and filtration. Total malondialdehyde (MDA) contents in extracts and plasma, an indicator of lipid oxidation, were measured using the thiobarbituric acid reactive substances (TBARS) method with spectrophotometry at 532 nm, as described by Grotto et al. [20]. GPx activities in extracts and plasma, defined as the amount of sample required to oxidize 1 μmol of 2,4-dinitrophenylhydrazine (DNPH) per min at 25 °C, were quantified by monitoring changes in NADPH oxidation using a Multi-Mode Microplate Reader at 340 nm [21]. SOD activity in bone marrow, expressed as the percentage inhibition of pyrogallol autoxidation, was determined by measuring absorbance changes at 420 nm over 5 min using spectrophotometry. Plasma SOD level was measured using a commercial kit (Nanjing Jiancheng Bioengineering Institute, Nanjing, China). To quantify glutathione (GSH) and glutathione disulfide (GSSG), erythrocytes were harvested by centrifugation (3000× *g*, 15 min), lysed with 70% metaphosphoric acid, and the lysate was collected. GSH and GSSG concentrations were determined via high-performance liquid chromatography, using γ-glutamyl-glutamate as an internal standard, iodoacetic acid as a thiol-quenching agent, and 1-chloro-2,4-dinitrobenzene as a derivatization reagent. Separation was performed on an aminopropyl column with absorbance measured at 365 nm. GSH and GSSG concentrations were normalized to protein content, and the GSSG/GSH ratio was calculated as previously described [22].

### 2.8. Measurement of Cytokine Levels

To evaluate the effect of MSM on inflammatory response in aged mice, pro-inflammatory cytokines, including interleukin (IL)-1β, IL-6, and tumor necrosis factor-alpha (TNF-α), and the anti-inflammatory cytokine IL-10 were measured using ELISA kits according to the manufacturer’s protocols. All kits were purchased from R&D Systems, Inc. (Minneapolis, MN, USA). 

### 2.9. Gene Expression Analysis

The proximal tibia and bone marrow samples were ground in liquid nitrogen and mixed with Trizol reagent (Sigma-Aldrich) to extract total RNA. After assessing RNA quality and concentration, the extracted RNA was reverse-transcribed into cDNA following the kit instructions. A qPCR system was established using both glyceraldehyde-3-phosphate dehydrogenase (GAPDH) and β-actin as the internal reference. The reaction conditions were set as follows: 95 °C for 30 s, followed by 40 cycles of 95 °C for 5 s and 60 °C for 34 s. Primers were designed using online Primer 3 website (https://primer3.ut.ee/, accessed on 1 December 2023) and shown in Appendix A. The mRNA expressions of target genes were normalized using housekeeping gene β-actin [23].

### 2.10. Statistical Analysis

The data from the present study were analyzed using IBM SPSS Statistics software for Windows Version 22.0 (SPSS Inc., USA) and were presented as mean ± standard deviation (SD). With a minimum effect size of 1.0, a statistical power exceeding 0.80 was achieved. The Shapiro–Wilk test was used to check the normality of the data distribution, and Levene’s test was applied to assess the homogeneity of variance. One-way analysis of variance (ANOVA), followed by Tukey’s post hoc test, was performed to evaluate the age-related differences in bone parameters and physiological indices among young, adult, and aged male mice. Comparisons between aged mice and aged mice treated with MSM were conducted using a two-tailed unpaired t-test. Additionally, Pearson’s correlation analysis was used to examine relationships between the parameters of bone mass and oxidative status, while survival rate was assessed using the Kaplan–Meier method. To define the relationship between body weight and age, logistic analysis was conducted by GraphPad Prism (Version 8.0, GraphPad Software Inc., CA, USA). Statistical significance was defined as *p* < 0.05.

## 3. Results

### 3.1. Bone Loss Was Caused by Aging in Rodent Model

As shown in Figure 1A, the mice were categorized into three age groups, i.e., young (8 weeks), adult (32 weeks) and aged (72 weeks). The body weight increased with age, with significant growth occurring before 22 weeks of age. The growth curve was modeled as follows: body weight = 39.79 × 3.18/(26.61 × Exp(−1.33 × week) +13.18) (R^2^ = 0.952; Figure 1B). The µCT analysis of the tibia revealed that the aged mice exhibited significantly greater bone loss compared to the young and adult mice (Figure 1C), as indicated by the reduced BV/TV and Tb.N values (*p* < 0.05; Figure 1D,E). No significant difference in the Tb.Th value was observed among the three age groups (Figure 1F). However, compared with the young mice, the aged mice displayed a decreased tibia Tb.Cd value and increased tibia Tb.Sp value (*p* < 0.05; Figure 1G,H), both of which were comparable to the adult group.

### 3.2. Aging Induced Oxidative Stress and Was Related to Bone Loss

The results of the oxidative status revealed that the plasma MDA level was significantly higher in the aged mice compared to the young and adult mice (*p* < 0.05; Figure 2A). Additionally, the activities of plasma SOD and GPx were decreased in the aged mice relative to those in the young mice (*p* < 0.05; Figure 2B,C). As shown in Figure 2D and Appendix A, the correlations between the bone characteristics and oxidative status indicated that the plasma MDA level was negatively correlated with the tibia BV/TV, Tb.N, and Tb.Cd values, but was positively correlated with the tibia Tb.Sp value. Conversely, the plasma SOD activity exhibited a positive correlation with the tibia BV/TV and Tb.Cd values and a negative correlation with the tibia Tb.Sp value. Furthermore, the plasma GPx activity was positively correlated with the tibia Tb.N value.

### 3.3. MSM Supplementation Reversed Age-Related Bone Loss

The aged mice treated with MSM exhibited no significant effects in terms of the body weight, feed intake, water consumption or survival proportions (Appendix A). However, the MSM supplementation significantly increased the max. load of the tibia (*p* < 0.05), which was accompanied by no difference in the stiffness (Figure 3A,B). Additionally, drinking water containing MSM improved the BV/TV and Tb.Cd values in the proximal tibia (*p* < 0.05), although it did not affect the Tb.N, Tb.Th, or Tb.Sp values in this trial (Figure 3D–H). In the cortical bone, the MSM treatment did not alter the Ct.Ar value (Figure 3I) but significantly improved the Ct.Th and Ct.BMD value (*p* < 0.05; Figure 3J,K).

### 3.4. Improved Bone Mass by MSM Treatment Was Linked to Reduced Bone Resorption

The MSM supplementation in the aged mice resulted in a significant decrease in the osteoclast frequency (TRAP-positive cells) observed in the bone sections (*p* < 0.05; Figure 4A,B). Additionally, the aged mice treated with MSM exhibited a lower concentration of CTx, a marker of bone resorption, in the serum (*p* < 0.05; Figure 4C). Regarding osteoclastogenic factors, the mRNA expression of nuclear factor of activated T-cells, cytoplasmic, calcineurin dependent 1 (*NFATc1*) and receptor activator of nuclear factor-κB ligand (*RANKL*) was significantly downregulated in the MSM-treated aged mice, leading to a reduced *RANKL*/Osteoprotegerin (*OPG*) ratio in the tibia (*p* < 0.05; Figure 4D). Furthermore, the MSM administration suppressed the expression of osteoclastic activity-related genes, including matrix metallopeptidase 9 (*MMP9*) (*p* < 0.05; Figure 4D) and cathepsin K (*Ctsk*) (*p* = 0.053; Figure 4D), compared to the aged mice receiving water without MSM. For serum markers and genes related to bone formation, MSM supplementation showed a trend toward an increased serum OCN level (*p* = 0.079; Figure 4F) and runt related transcription factor 2 (*Runx2*) mRNA abundance (*p* = 0.084; Figure 4G). Moreover, significant increases were observed in osteocyte-related genes, including sclerostin (*Sost*) and collagen type I alpha 1 (*Col1a1*), in MSM-treated aged mice compared to the untreated aged group (*p* < 0.05; Figure 4G).

### 3.5. MSM Manipulation Promoted Antioxidative Capability in Aged Mice

The effects of MSM supplementation on the antioxidative status are summarized in Table 1 and Figure 5. While no significant changes were observed in the plasma MDA content or SOD activity, the MSM treatment significantly increased the GPx activity in the plasma of the aged mice (*p* < 0.05; Table 1). Additionally, the MSM supplementation elevated the GSH content (*p* < 0.05; Table 1) and showed a trend toward reducing the GSSG/GSH ratio in erythrocytes (*p* = 0.072; Table 1). In the bone marrow, the MSM supplementation increased the GPx activity and GSH content, and decreased the MDA level and the GSSG/GSH ratio (*p* < 0.05; Table 1). Regarding the expression of oxidative stress-related genes, the MSM treatment upregulated the mRNA expression of nuclear factor E2 related factor 2 (*Nrf2*) and *GPx1* (*p* < 0.05; Figure 5A,F) and tended to increase the mRNA levels of *SOD1* (*p* = 0.060; Figure 5C) and *SOD2* (*p* = 0.061; Figure 5C) in the bone marrow of the aged mice. However, the mRNA levels of heme oxygenase 1 (*HO-1*), catalase (*CAT*), and glutathione reductase (*GR*) in the bone marrow were not remarkably altered by the MSM addition (Figure 5B,D,E).

### 3.6. MSM InclusionReduced Inflammatory Status in Aged Mice

As illustrated in Figure 6A,C, the MSM supplementation significantly reduced the serum concentrations of pro-inflammatory cytokines IL-1β and TNF-α in the aged mice compared to the untreated aged group (*p* < 0.05). However, no notable differences were observed in the serum levels of pro-inflammatory cytokine IL-6 or anti-inflammatory cytokine IL-10 between the aged mice with and without the MSM treatment (Figure 6B,D). In addition, the mRNA expression of pro-inflammatory genes (Figure 6E), including nuclear factor-kappa B (*NF-κB*), *TNF-α*, *IL-6*, and *IL-1β*, was downregulated to different extents in response to the MSM supplementation in the aged mice.

## 4. Discussion

Age-related bone fragility is a significant public health concern and is extensively studied, often linked to abnormal behavior and pain. In this study, we characterized changes in murine bone associated with natural aging, with a focus on alterations in the tibial microstructure. Age-induced bone loss was evident, consistent with bone-related diseases such as osteoporosis, which involve microstructural damage and reduced bone mass [1]. The aged mice demonstrated reduced bone formation, indicated by increased Tb.Sp and decreased BV/TV, Tb.N, Tb.Th, bone mineral content, and bone mineral density compared to the younger mice [24]. Notably, aging is frequently accompanied by oxidative stress [3], suggesting a role for oxidative substances in bone loss. In this study, aged mice showed an elevated MDA level and reduced SOD and GPx activities in plasma, indicating heightened oxidative stress. The correlation analysis revealed negative relationships between the plasma MDA level and bone characteristics (BV/TV, Tb.N, and Tb.Cd) and positive correlations between the SOD/GPx activities and bone metrics. These findings confirmed that oxidative stress contributed to age-related bone loss. Supporting evidence has been reported showing that *SOD1* or *FoxO1,3,4* knockout mice exhibit increased ROS and reduced bone mass [7,25]. Additionally, the aging mice displayed decreased bone strength associated with elevated ROS and reduced GPx activity [8]. Importantly, antioxidant interventions such as vitamin C [26] and N-acetyl-L-cysteine [27] have been shown to mitigate oxidative stress-induced bone loss, highlighting potential therapeutic strategies.

Given the detrimental effects of oxidative substances on bone mass, MSM, characterized by antioxidant and anti-inflammatory properties [12,28], may mitigate age-induced bone loss. To investigate this, tibial scans revealed that the MSM supplementation improved the bone mass in the aged mice, as evidenced by increases in BV/TV, Tb.Cd, Ct.Th and Ct.BMD. These improvements contributed to an enhanced tibial maximum load capacity, aligning with the understanding that bone strength depends primarily on trabecular microstructure and cortical composition [29]. Similar findings were reported in a previous study, where MSM-treated aging mice showed an improved trabecular microstructure, including increased BV/TV, Tb.Th, and Tb.N, along with decreased Tb.Sp [30]. Mechanistically, osteocytes, the predominant cell type in the mineralized bone matrix, form a complex network that functions as the primary sensor of mechanical loads and a regulator of mineral homeostasis [31,32]. Aging disrupted this network by reducing osteocyte numbers, increasing senescent osteocytes, and impairing their ability to sense mechanical loads [32]. In this study, the MSM treatment enhanced the mRNA expression of osteocyte-specific genes such as Sost and Col1a1, further supporting the positive effects of MSM on bone health. Taken together, these findings demonstrated that supplementation with 400 mg/kg/d MSM in drinking water could effectively alleviate age-induced bone loss.

Bone quality is strongly influenced by the balance between bone formation and resorption. Emerging evidence suggests that MSM enhances the mRNA expression of *Runx2* in mesenchymal stem cells and stem cells derived from human exfoliated deciduous teeth [15,16]. However, in this study, the inclusion of MSM in drinking water only slightly increased the *Runx2* mRNA expression and serum OCN level, a marker of bone formation, in the aged mice. These findings indicate that MSM’s effect on bone formation in vivo is limited. Nevertheless, in an aging mouse model, subcutaneous MSM-injected mice (100 mg/kg body weight) presented an increase in the bone mass of the mandibular, which was attributed to increased bone formation markers [30]. This implied that the roles of MSM in age-related bone loss must be further evaluated in a future study. Regarding bone resorption, the aged mice treated with MSM exhibited reduced osteoclast-mediated bone resorption, as evidenced by the decreased serum CTx level, a marker of bone resorption. This reduction was further supported by a lower number of osteoclasts (N.Oc/B.Pm) and decreased activities of matrix-degrading enzymes, including matrix MMP9 and Ctsk, which were secreted by osteoclasts to degrade the organic bone matrix [33]. It is well established that osteoclastogenesis is driven by the interaction of RANKL with its receptor RANK, leading to osteoclast differentiation [34]. The overexpression of RANKL accelerated osteoclastogenesis, resulting in osteoporosis, while RANKL deletion caused osteopetrosis due to osteoclast deficiency [35,36]. OPG, an osteoclastogenesis inhibitory factor, binds to RANKL and negatively regulates osteoclast differentiation [37]. Additionally, NFATc1, a key transcription factor, is activated by RANKL-RANK binding, promoting the expression of osteoclastic genes like *MMP9* and *Ctsk*, thereby driving osteoclast differentiation [33]. In the present study, MSM suppressed osteoclastogenesis and bone resorption by downregulating the mRNA expression of *NFATc1*, *RANKL*, *MMP9*, and *Ctsk*, and the *RANKL/OPG* ratio. These findings were consistent with the in vitro studies, which showed that MSM inhibited RANKL-stimulated osteoclast differentiation in bone marrow macrophages by suppressing the mRNA expression of *NFATc1* and *Ctsk* [38]. Overall, the results suggested that MSM improved the bone mass in aged mice primarily by suppressing bone resorption rather than enhancing bone formation.

To explore the potential mechanism by which MSM suppresses bone resorption, its antioxidative capability is considered to be a key factor. ROS play a critical role in osteoclast differentiation [39]. Excessive osteoclast activity may be mitigated by reducing intracellular ROS production [40]. Given the link between diminished SOD and GPx activities and bone loss in aged mice, the levels of these antioxidant enzymes were evaluated in this study. Consistent with recent findings [41], the MSM supplementation did not alter the plasma or bone marrow SOD activity in the aged mice. However, the reduction in the MDA level in the bone marrow suggested an improved antioxidant status due to MSM, likely mediated by increased GPx activity. GPx detoxifies hydrogen peroxide to water using GSH as a substrate [42], while oxidized GSH (GSSG) is regenerated to GSH by GR [43]. Thus, the redox balance, measured by GSH, GSSG, and their ratio, is a crucial indicator of the antioxidant capacity. MSM metabolism supplies organic sulfur, a precursor for GSH synthesis, counteracting GSH depletion [4]. In this study, MSM increased the GSH and GPx levels, decreased the GSSG/GSH ratio, and enhanced the antioxidant status of the aged mice. These results align with prior research in rats and broilers [28,41,44]. However, some studies have reported no improvement in the GSH level or GSSG/GSH ratio with MSM supplementation [45,46], possibly due to differences in experimental models, MSM dosage, or treatment duration. As far as the dose of MSM used in this study (400 mg/kg/d) is concerned, there were multiple doses in previous studies to exert its biological function, such as an oral gavage of 1500 mg/kg BW daily for 21 d, which did not cause any adverse effects on growth or clinical outcomes and appeared to be absorbed and distributed throughout the body [47], indicating the optimal dose of MSM for maximizing bone mass required in the current study. Nrf2, a key transcription factor for oxidative stress response, regulates the expression of cytoprotective and antioxidant enzymes, supporting glutathione and thioredoxin systems to mitigate oxidative damage [48,49,50]. Moreover, sulfonyl-containing compounds have been confirmed to be involved in the suppression of the Keap1-Nrf2 interaction and activation of the Nrf2 pathway [51]. Herein, in this study, the MSM treatment increased the mRNA expression of *Nrf2* and *GPx*, as well as upregulated *SOD1* and *SOD2* transcription in bone marrow, consistent with previous findings [17,52]. These findings suggested that MSM improved the bone mass in aged mice, primarily by suppressing bone resorption, likely through its antioxidant properties.

The enhanced antioxidant defense induced by the Nrf2 pathway can suppress the transcriptional activity of NF-κB, thereby mitigating the inflammatory response [50]. This is particularly relevant in aging, as chronic inflammation impairs bone regeneration and contributes to bone loss [53]. For example, TNF-α, secreted along with IL-1 by mononuclear cells, indirectly promoted osteoclastogenesis by stimulating RANKL expression and facilitating the binding of RANKL to its receptor RANK on osteoclast precursors [54]. TNF-α treatment significantly increased circulating osteoclast precursors, an effect reversed by anti-TNF-α therapy in mice [55]. Similarly, IL-1β enhanced osteoclastogenesis by increasing the *RANKL* mRNA level and the *RANKL/OPG* ratio [56]. Herein, upregulated RANKL expression could promote the activation of NF-κB and then stimulate the initial expression of NFATc1, thereby enhancing osteoclastogenesis [33]. Previous studies have shown that MSM reduced inflammatory responses by decreasing the TNF-α expression in LPS-stimulated RAW264.7 cells [13], and attenuating TNF-α-induced inflammation in cardiac cells [46]. MSM administration has also been demonstrated to suppress the transcriptional expression of IL-1β [57] and mitigate the adverse effects of IL-1β on cell viability via the NF-κB signaling pathway [58]. In the present study, MSM supplementation effectively reduced inflammatory responses in aged mice. This was manifested by decreased serum levels of pro-inflammatory cytokines, including TNF-α and IL-1β, as well as downregulated mRNA expression of *TNF-α*, *IL-1β* and *NF-κB* in bone marrow. These results suggested that MSM supplementation in drinking water inhibited the RANKL/NF-κB/NFATc1 signaling pathway, thereby suppressing osteoclast differentiation, consistent with findings from a previous study [38].

## 5. Conclusions

In conclusion, a limitation of this study was the age categorization of the mice, defined as young (8 weeks), adult (32 weeks) and aged (72 weeks). It is well established that bone formation and resorption dynamics vary significantly across different life stages. Herein, including a second group of younger aged mice (50–60 weeks old) in the study would make the findings more convincing, as this may better demonstrate the role of MSM and provide evidence that its preventive effects on bone loss precede clinical symptoms, rather than serving solely as a treatment for age-related bone loss. In addition, the age-related loss of bone mass and strength is a common skeletal feature influenced by multiple factors, including hormonal changes (e.g., estrogen deficiency), nutritional deficiencies (e.g., calcium and vitamin D), increased oxidative stress from ROS and reduced physical activity [59]. Notably, estrogen plays a crucial role in bone homeostasis by inhibiting osteoclast activity and promoting osteoblast survival. In females, estrogen decline (e.g., postmenopause) leads to increased bone resorption, making them more susceptible to osteoporosis and bone loss. Additionally, males typically have greater muscle mass and higher mechanical loading on bones, which enhances the bone density through mechanotransduction. Therefore, sex differences should be considered, and further studies are needed to clarify the role of MSM in female models. This study primarily investigated the relationship between age-induced bone loss and oxidative stress using correlation analysis and MSM supplementation. While the findings confirmed a significant link between oxidative stress and bone loss, further studies are necessary to elucidate the underlying mechanisms. Within these limitations, we demonstrated that the MSM supplementation effectively improved the bone mass in the aged mice, primarily by regulating bone resorption. The antioxidant properties of MSM likely played a key role in this suppression of bone resorption.

## Figures and Tables

**Figure 1 antioxidants-14-00216-f001:**
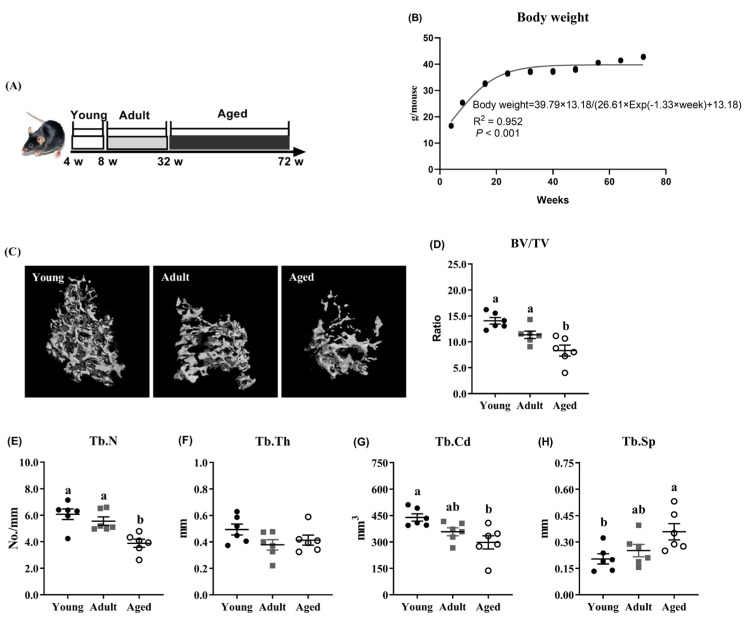
Aging-induced bone loss in mice. (**A**) Mice were categorized as young (8 weeks), adult (32 weeks) and aged (72 weeks). (**B**) Body weight. (**C**) Micro-CT images of tibia trabecular bone. Tibia trabecular bone parameters including (**D**) the ratio of bone volume to total volume (BV/TV), (**E**) trabecular number (Tb.N), (**F**) trabecular thickness (Tb.Th), (**G**) trabecular connectivity density (Tb.Cd), and (**H**) trabecular spaces (Tb.Sp). ^a,b^ Values (mean ± standard deviation; *n* = 6) with different letters represent statistical significance at *p* < 0.05.

**Figure 2 antioxidants-14-00216-f002:**
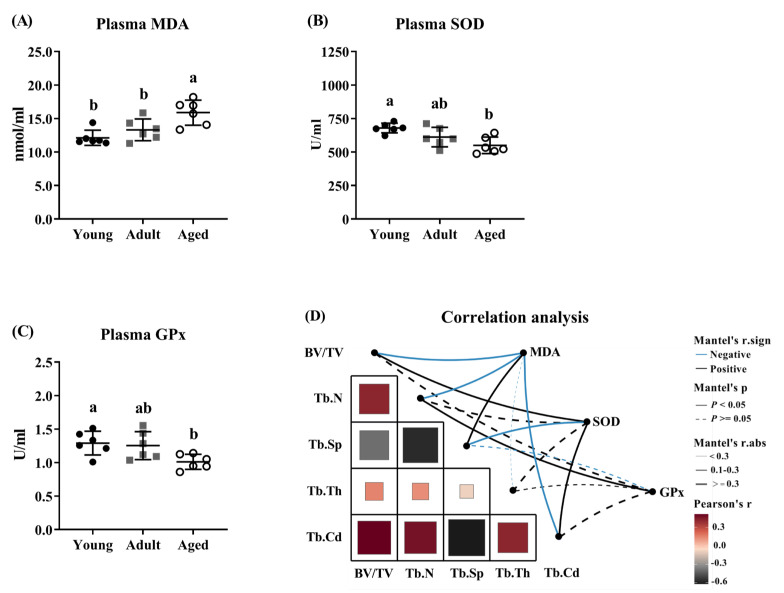
Age-induced bone loss was related to oxidative stress in mice. Plasma oxidative stress parameters of (**A**) malondialdehyde (MDA), (**B**) superoxide dismutase (SOD) and (**C**) glutathione peroxidase (GPx). (**D**) Correlations between bone characteristics and plasma oxidative status. ^a,b^ Values (mean ± standard deviation; *n* = 6) with different letters represent statistical significance at *p* < 0.05. In this study, mice were categorized as young (8 weeks), adult (32 weeks) and aged (72 weeks). BV/TV, the ratio of bone volume to total volume; Tb.N, trabecular number; Tb.Sp, trabecular spaces; Tb.Th, trabecular thickness; Tb.Cd, trabecular connectivity density.

**Figure 3 antioxidants-14-00216-f003:**
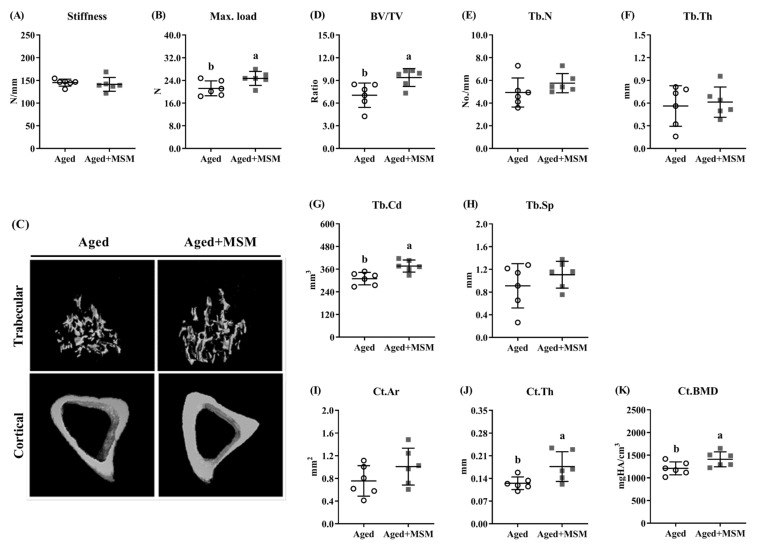
MSM supplementation improved tibia bone mass in aged mice. The mechanical properties were determined including (**A**) stiffness (the slope of the linear portion of the load–displacement curve) and (**B**) the maximum force (max. load, Newtons). Tibia was subjected to scans using (**C**) μCT and bone mass was quantified, including (**D**) the ratio of bone volume to total volume (BV/TV), (**E**) trabecular number (Tb.N), (**F**) trabecular thickness (Tb.Th), (**G**) trabecular connectivity density (Tb.Cd), and (**H**) trabecular spaces (Tb.Sp). Tibia cortical bone parameters included (**I**) cortical area (Ct.Ar), (**J**) cortical thickness (Ct.Th), and (**K**) cortical bone mineral density (Ct.BMD). ^a,b^ Values (mean ± standard deviation; *n* = 6) with different letters represent statistical significance at *p* < 0.05.

**Figure 4 antioxidants-14-00216-f004:**
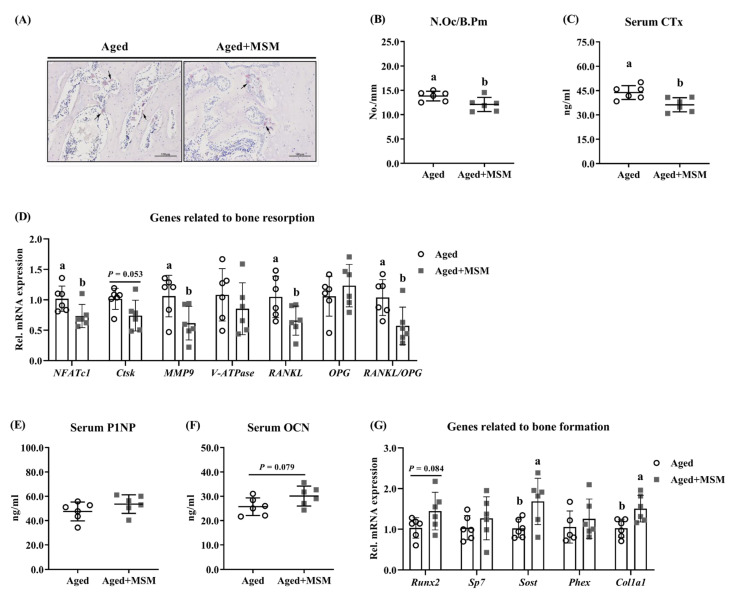
The alterations of aged mice responses to MSM supplementation in drinking water. (**A**) Tartrate-resistant acid phosphatase (TRAP) staining, in which the positive-TRAP cells (osteoclast) were indicated by the black arrow, and (**B**) the frequency of osteoclasts (N.Oc/B.Pm) were determined. (**C**) Serum C-terminal telopeptide of type I collagen (CTx) content reflects bone resorption. (**D**) The mRNA expression of genes related to bone resorption in tibia, including nuclear factor of activated T-cells, cytoplasmic, calcineurin dependent 1 (*NFATc1*), receptor activator of nuclear factor-κB ligand (*RANKL*), osteoprotegerin (*OPG*), cathepsin K (*Ctsk*), matrix metallopeptidase 9 (*MMP9*), and vacuolar-type H^+^-ATPase (*V-ATPase*). Serum bone formation biomarkers such as (**E**) N-terminal propeptide of procollagen type I (P1NP) and (**F**) osteocalcin (OCN). (**G**) The mRNA expression of genes related to bone formation in tibia, including runt related transcription factor 2 (*Runx2*), Sp7 transcription factor 7 (*Sp7*), sclerostin (*Sost*), phosphate regulating endopeptidase homolog x-linked (*Phex*), and collagen type I alpha 1 (*Col1a1*). ^a,b^ Values (mean ± standard deviation; *n* = 6) with different letters represent statistical significance at *p* < 0.05.

**Figure 5 antioxidants-14-00216-f005:**
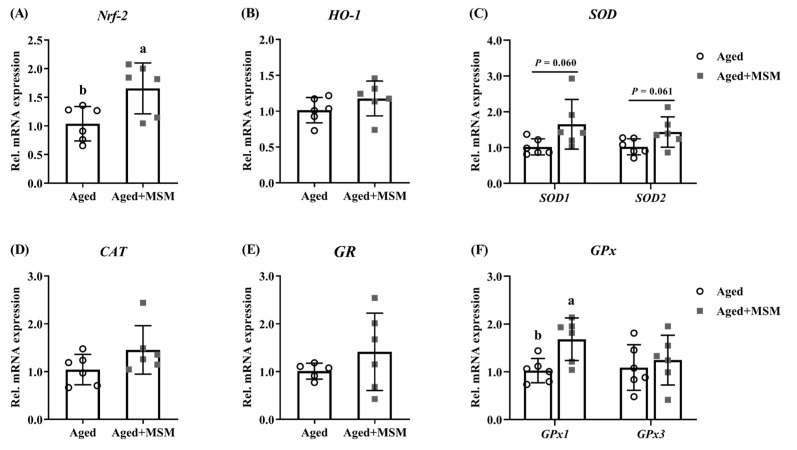
Impacts of MSM supplementation on the mRNA expression of genes related to antioxidative status in bone marrow. (**A**) Nuclear factor E2 related factor 2 (*Nrf2*), (**B**) heme oxygenase 1 (*HO-1*), (**C**) superoxide dismutase (*SOD*), (**D**) catalase (*CAT*), (**E**) glutathione reductase (*GR*), and (**F**) glutathione peroxidase (*GPx*). ^a,b^ Values (mean ± standard deviation; *n* = 6) with different letters represent statistical significance at *p* < 0.05.

**Figure 6 antioxidants-14-00216-f006:**
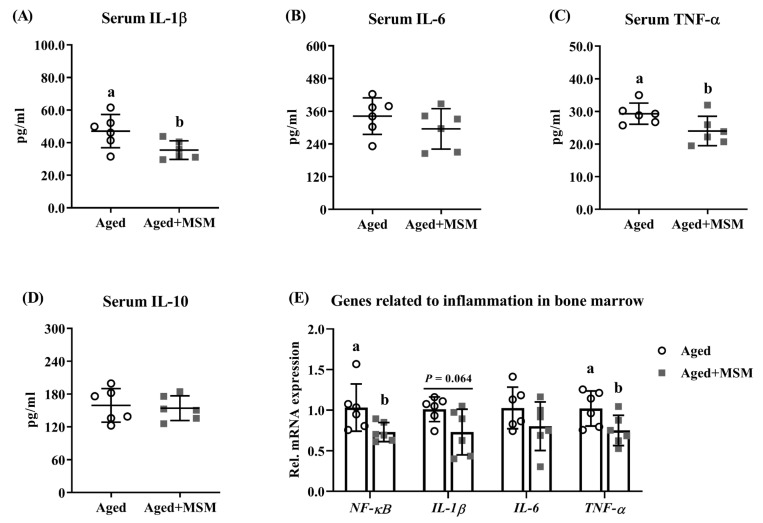
MSM supplementation in drinking water suppresses inflammatory response in aged mice. The concentrations of inflammatory cytokines such as (**A**) interleukin-1 beta (IL-1β), (**B**) interleukin-6 (IL-6), (**C**) tumor necrosis factor-alpha (TNF-α), and (**D**) interleukin-10 (IL-10). (**E**) The mRNA expression of genes related to inflammation in bone marrow, including nuclear factor-kappa B (*NF-κB*), *IL-1β*, *IL-6*, *TNF-α*. ^a,b^ Values (mean ± standard deviation; *n* = 6) with different letters represent statistical significance at *p* < 0.05.

**Table 1 antioxidants-14-00216-t001:** Effects of MSM supplementation on antioxidative status in aged mice.

Items	Aged	Aged + MSM	*p*-Value
Plasma			
MDA (nmol/mL)	19.27 ± 2.62	17.74 ± 1.34	0.231
SOD (U/mL)	483.37 ± 75.54	565.25 ± 90.36	0.119
GPx (U/mL)	1.04 ± 0.12 ^b^	1.19 ± 0.10 ^a^	0.044
Erythrocytes			
GSH (μmol/mL)	0.752 ± 0.173 ^b^	1.017 ± 0.221 ^a^	0.043
GSSG (μmol/mL)	0.013 ± 0.001	0.011 ± 0.006	0.599
GSSG/GSH	0.017 ± 0.002	0.011 ± 0.006	0.072
Bone marrow			
MDA (nmol/g)	18.98 ± 2.03 ^a^	16.19 ± 2.20 ^b^	0.045
SOD (U/g)	14.24 ± 1.47	14.48 ± 2.87	0.857
GPx (U/g)	4.51 ± 0.64 ^b^	5.47 ± 0.49 ^a^	0.015
GSH (μmol/g)	1.215 ± 0.55 ^b^	2.192 ± 0.314 ^a^	0.004
GSSG (μmol/g)	0.032 ± 0.007	0.028 ± 0.013	0.502
GSSG/GSH	0.033 ± 0.018 ^a^	0.013 ± 0.005 ^b^	0.040

^a,b^ Values (mean ± standard deviation; *n* = 6) with different letters in the same row represent statistical significance at *p* < 0.05. MDA, malondialdehyde; SOD, superoxide dismutase; GPx, glutathione peroxidase; GSH, glutathione; GSSG, glutathione disulfide.

## Data Availability

Data will be made available on request.

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
