# Peer review of "Protective Effect of Methyl Sulfonyl Methane on the Progression of Age-Induced Bone Loss by Regulating Oxidative Stress-Mediated Bone Resorption"

_antioxidants, 2025, doi:10.3390/antiox14020216_

Round 1

Reviewer 1 Report

This research investigated the effect of methylsulfonylmethane (MSM) on the progression of age-induced bone loss including the anti-oxidative effects. The contents of this article are in line with the previous studies and will provide scientific insight into this research field.

Major concerns:

The results and contents of this article may be sufficient to support the claim of this study, whereas the way MSM dosage was decided is not described in detail. The methods state that 5% (v/v) MSM was adopted based on the previous articles, in which many studies utilized 5% or similar dosages, the more high concentrations can be evaluated in the further research. The authors described this limitation in the discussion section (L392), but the specific discussion of the concentration (i.e. 5% or more) can be added to this section.

Minor concerns:

L109 marrow, and

L145 erythrocyte, and

Author Response

The results and contents of this article may be sufficient to support the claim of this study, whereas the way MSM dosage was decided is not described in detail. The methods state that 5% (v/v) MSM was adopted based on the previous articles, in which many studies utilized 5% or similar dosages, the more high concentrations can be evaluated in the further research. The authors described this limitation in the discussion section (L392), but the specific discussion of the concentration (i.e. 5% or more) can be added to this section.

Thank you for your professional suggestions, we improved the description as follows: “However, some studies reported no improvement in GSH level or GSSG/GSH ratio with MSM supplementation [45,46], possibly due to differences in experimental models, MSM dosage, or treatment duration. As far as the dose of MSM used in this study (5%, v/v) is concerned, there were multiple doses in previous studies to exert its biological function, such as oral gavage of 1,500 mg/kg BW daily for 21 d) did not cause any adverse effects on growth or clinical outcomes and appeared to be absorbed and distributed throughout the body [47], indicating the optimal dose of MSM for maximizing bone mass required to explored in the following study”. Thanks.

 Minor concerns:

L109 marrow, and

We revised. Thanks.

L145 erythrocyte, and

We revised. Thanks.

Reviewer 2 Report

The article describes the use of MSM (methyl sulfonyl methane) as a dietary supplement to prevent age-associated bone loss in a mouse aged model. They test the hypothesis that MSM, which has shown antioxidant activity and immune modulation in rat models, can ameliorate bone loss in aged mice, assuming that aged-associated bone loss is linked to an imbalance in the redox status of the organism, driving to a pro-inflammatory stage and altered bone resorption vs. bone formation, leading finally to a decrease in bone mass. 

The authors define three experimental groups of animals: young, adult and aged, and study multiple parameters related to bone quality (cortical and trabecular bone mass, biomechanics, bone formation/resorption dynamics) and to oxidative and inflammatory status. They find a correlation between mice age, increased oxidative stress, proinflammatory status and decreased bone quality due to increased resorption. Although this correlation has been previously described, the authors hereby provide new strong experimental evidence in the murine model, based on a very complete and exhaustive set of assays.

Authors then offer aged mice an MSM supplement in their drinking water (controls receiving the same diet without the supplement) and repeat the studies, finding that MDM increases some bone quality-related parameters, particularly biomechanics loading, which is a functional important parameter. They also find decreased bone resorption as compared to controls, which may account for the better loading performance rather than increased bone formation, which did not occur. A reduced oxidative and inflammatory status was found in MDM-treated mice as compared to controls. Given the connection between ROS and osteoclast differentiation, and that of inflammatory cytokines and osteclastogenesis and osteoclast activation, the authors offer a mechanistic explanation for MDM effects on bone. They admit that confirmation of this explanation will require further experimentation. However, they prove that MDM supplementation in an aged mice model regulates oxidative and inflammatory status, and effectively improves bone mass and biomechanics, through a downregulation in bone resorption.

Bone loss in elders is a main concern for Health Systems as it drives elder individuals to impaired mobility and reduced self-sufficiency. It also is related to other osteochondral and locomotor diseases that require medication, physiotherapy and other medical attention, all of them representing a load on National Health Systems. For this reason, research of new drugs that prevent or reduce age-related bone loss is becoming more and more important. In this context, the article provides positive evidence on the potential positive use of MDM to prevent age-associated bone mass.

The article is well written, with a clear and well documented introduction, exhaustively described materials and methods, and well presented and illustrated data. The discussion relates their results to those of previous studies, many in rat models, and proposes a potential mechanism connecting redox species, inflammatory cytokines and bone quality/health; and how MDM supplementation in diet impacts this mechanisms to improve bone health. The authors are honest as to the need of further experimental evidence to support the proposed mechanism, but they prove their point as to the correlation between the mentioned parameters and the positive effects of MDM supplementation in aged-mice bone quality.

I wonder just two points:

  • Based on water consumption and mice weight, authors could have estimated the average daily dose of MDM ingested by the animals, and compare it to the dose received in previous experiments, particularly those where MDM was injected. Maybe in this study the dosing of MDM is low and a higher dose could have yielded better, more “spectacular” results.
  • The age chosen by the authors to test MDM supplementation (73 week old) is alright, and they got an effect, but these mice were really old. It would have been interesting to include a second group of aged mice, younger than the first (between 50 to 60 week old) in the study. Possibly in this group of “younger elders” MDM effects would have been more radical. It is becoming increasingly evident that healthy aging is based on prevention rather than treatment. Proving that benefits of MDM are more pronounced at the beginning of the elder age would be a point supporting that “healthy aging should start as young as possible”. 

Author Response

I wonder just two points:

Based on water consumption and mice weight, authors could have estimated the average daily dose of MSM ingested by the animals and compared it to the dose received in previous experiments, particularly those where MSM was injected. Maybe in this study the dosing of MSM is low, and a higher dose could have yielded better, more “spectacular” results.

Thank you for revieing carefully and presenting professional suggestions.

Indeed, there are multiple doses in previous studies to exert its biological function, such as oral gavage of 1,500 mg/kg BW daily for 21 d did not cause any adverse effects on growth or clinical outcomes and appeared to be absorbed and distributed throughout the body [47]. We total agree with your suggestion, the dosing of MDM is low, and a higher dose could have yielded better, more “spectacular” results. This also indicates the optimal dose of MSM for maximizing bone mass required to explore in the following study. We will continue to evaluate the dose of MSM used in bone quality using different animal models. Therefore, we point out that the limitations in manuscript are as follows:

“However, some studies reported no improvement in GSH level or GSSG/GSH ratio with MSM supplementation [45,46], possibly due to differences in experimental models, MSM dosage, or treatment duration. As far as the dose of MSM used in this study (5%, v/v) is concerned, there were multiple doses in previous studies to exert its biological function, such as oral gavage of 1,500 mg/kg BW daily for 21 d did not cause any adverse effects on growth or clinical outcomes and appeared to be absorbed and distributed throughout the body [47], indicating the optimal dose of MSM for maximizing bone mass required to explored in the following study”.

Thanks, again.

The age chosen by the authors to test MDM supplementation (73-week-old) is alright, and they got an effect, but these mice were really old. It would have been interesting to include a second group of aged mice, younger than the first (between 50- to 60-week-old) in the study. Possibly in this group of “younger elders” MDM effects would have been more radical. It is becoming increasingly evident that healthy aging is based on prevention rather than treatment. Proving that benefits of MDM are more pronounced at the beginning of the elder age would be a point supporting that “healthy aging should start as young as possible”.

We are truly grateful for your critical comments.

It is true, including a second group of younger aged mice (50–60 weeks old) in the study would make the findings more convincing, as it may better demonstrate the role of MSM and provide evidence that its preventive effects on bone loss precede clinical symptoms, rather than serving solely as a treatment for age-related bone loss.

According to your suggestions, we added the related content in this limitation section as follows: “a limitation of this study was the age categorization of mice, defined as young (8 weeks), adult (32 weeks), and aged (72 weeks). It is well-established that bone formation and resorption dynamics vary significantly across different life stages. Herein, including a second group of younger aged mice (50–60 weeks old) in the study would make the findings more convincing, as it may better demonstrate the role of MSM and provide evidence that its preventive effects on bone loss precede clinical symptoms, rather than serving solely as a treatment for age-related bone loss”

Therefore, the referee’s concern is of importance for our further study, and we will show the results in our next paper. Here, we seek for the reviewer and editor’s tolerance and understanding. Many thanks for your kind help!

Reviewer 3 Report

This manuscript studied bone metabolism and oxidation status according to age differences in young, adult, and old rats. The type of experiment is simple and very reasonable strategy, but the results and suggestions presented are very interesting. Therefore, this manuscript is worthy of the journal publication.

Genes related to bone metabolism and oxidation are used to be regulated differently in females and males, and estrogen-related pathways are also important, but only male rats were used in this study. Therefore, I would like to suggest additional explanations for sex differences or females in the discussion section

Author Response

This manuscript studied bone metabolism and oxidation status according to age differences in young, adult, and old rats. The type of experiment is simple and very reasonable strategy, but the results and suggestions presented are very interesting. Therefore, this manuscript is worthy of journal publication.

We appreciate the reviewer’s comments.

Genes related to bone metabolism and oxidation are used to be regulated differently in females and males, and estrogen-related pathways are also important, but only male rats were used in this study. Therefore, I would like to suggest additional explanations for sex differences or females in the discussion section.

Regarding the referee’s concern, estrogen plays a crucial role in bone homeostasis by inhibiting osteoclast activity and promoting osteoblast survival. In females, estrogen decline (e.g., postmenopause) leads to increased bone resorption, making them more susceptible to osteoporosis and bone loss. Additionally, males typically have greater muscle mass and higher mechanical loading on bones, which enhances bone density through mechanotransduction. Therefore, sex differences should be considered, and further studies are needed to clarify the role of MSM in female models. For this, we specified the limitations in this study as follows:

“Notably, estrogen plays a crucial role in bone homeostasis by inhibiting osteoclast activity and promoting osteoblast survival. In females, estrogen decline (e.g., postmenopause) leads to increased bone resorption, making them more susceptible to osteoporosis and bone loss. Additionally, males typically have greater muscle mass and higher mechanical loading on bones, which enhances bone density through mechanotransduction. Therefore, sex differences should be considered, and further studies are needed to clarify the role of MSM in female models”.

Thanks again.